# Effects of fermented feeds and ginseng polysaccharides on the intestinal morphology and microbiota composition of Xuefeng black-bone chicken

Yueqin Xie⊙, Jie Liu⊙, Huan Wang, Junyi Luo, Ting Chen, Qianyun Xi, Yongliang Zhang*, Jiajie Sun ⬤ *

College of Animal Science, Guangdong Provincial Key Laboratory of Animal Nutrition Control, Guangdong Laboratory for Lingnan Modern Agriculture, South China Agricultural University, Guangzhou, Guangdong, China

⊙ These authors contributed equally to this work.
* zhangyl@scau.edu.cn (YZ); jiajiesun@scau.edu.cn (JS)

**Data Availability Statement:** All relevant data are within the paper and its Supporting Information files.

## Abstract

Fermented feeds contain abundant organic acids, amino acids, and small peptides, which improve the nutritional status as well as the morphology and microbiota composition of the intestine. Ginseng polysaccharides exhibit several biological activities and contribute to improving intestinal development. Here, Xuefeng black-bone chickens were fed a basal diet fermented by *Bacillus subtilis*, *Saccharomyces cerevisiae*, *Lactobacillus plantarum*, and *Enterococcus faecium*, with or without ginseng polysaccharides. The 100% microbially fermented feed (Fe) and 100% microbially fermented feed and ginseng polysaccharide (FP) groups showed significantly increased villus height and villus height to crypt depth ratio, and decreased crypt depth in the jejunum. In the 100% complete feed and ginseng polysaccharide (Po) group, the villus height to crypt depth ratio was significantly increased, crypt depth was reduced, and villus height remained unaffected. Next, we studied the intestinal microbial composition of 32 Xuefeng black-bone chickens. A total of 10 phyla and 442 genera were identified, among which *Firmicutes*, *Proteobacteria*, and *Bacteroidetes* were the most dominant phyla. At the genus level, *Sutterella* and *Asteroleplasma* abundance increased and decreased, respectively, in the FP and Po groups. *Sutterella* abundance was positively correlated to villus height and villus height to crypt depth ratio, and negatively correlated to crypt depth, and *Asteroleplasma* abundance was positively correlated to crypt depth and negatively correlated to villus height to crypt depth ratio. At the species level, the FP group showed significantly increased *Bacteroides_vulgatus* and *Eubacterium_tortuosum* and decreased *Mycoplasma_gallinarum* and *Asteroleplasma_anaerobium* abundance, and the Po group showed significantly increased *Mycoplasma_gallinarum* and *Asteroleplasma_a-naerobium* abundance. Moreover, bacterial abundance was closely related to the jejunum histomorphology. *Asteroleplasma_anaerobium* abundance was positively correlated with crypt depth and negatively correlated with villus height to crypt depth ratio. *Mycoplasma_-gallinarum* abundance was positively correlated to villus height, and *Bacteroides_vulgatus* and *Eubacterium_tortuosum* abundance was positively correlated with villus height to crypt

**Funding:** This study was supported by the Technical System of Modern Agricultural Industry in Guangdong Province [2018LM1121, 2018LM2158].

**Competing interests:** The authors have declared that no competing interests exist.

depth ratio and negatively correlated with crypt depth. Therefore, fermented feeds with ginseng polysaccharides may be used as effective alternatives to antibiotics for improving intestinal morphology and microbial composition.

## Introduction

It is well known that antibiotic growth promoters are added to animal feed for livestock growth promotion. However, the overuse of antibiotics has led to antibiotic resistance in animal microbial populations and raises the risk of transfer of antibiotic resistance genes to the human microbiota [1], thereby posing a threat to global public health. Hence, the use of antibiotics in animal feeds has been prohibited in many countries [2], which has led to a decline in animal production and an increase in the risk of food-borne infections in consumers due to higher rates of infections in livestock [3]. In order to solve the problems associated with the ban of antibiotics in livestock production, researchers are working towards finding new alternatives to antibiotics, such as the use of feed fermented by microbes. In general, the basal diet that fermented with probiotics has been extensively studied due to its benefits of increasing nutrient bioavailability and nutritional value [4]. Many researchers have reported that the use of probiotics for feed fermentation improves growth performance [4], meat quality [5], ileal amino acid digestibility [6], and immune function [7] in broilers. Other alternatives include addition of biological and functional additives to the diet, such as moringa, mulberry, Chinese herbal medicine, tea polyphenol, dietary fiber, and ginseng polysaccharides. Among them, ginseng polysaccharides, due to their high content of polysaccharides, peptides, saponins, and other active substances, can be used as functional feed additives to promote food intake [8], improve immunity [9], and decrease the amount of abdominal fat and serum cholesterol [10] in broilers. Therefore, if ginseng polysaccharide is mixed with microbially fermented feed, it may have a positive impact on the development of new alternative antibiotics. To date, however, no studies have focused on evaluating the effect of the combined use of these two substances in livestock and poultry diet. Xuefeng black-bone chicken is the most famous native breed in the Hunan province of China. It is greatly favored by people because its meat contains high contents of protein, vitamins, and amino acids [11], which has led to its increased demand. However, its slow growth, long feeding period, and risk of disease during the breeding process largely limits the development of the Xuefeng black-bone chicken industry. In the present study, we aimed to investigate the effect of feed fermented by microbes and ginseng polysaccharide on Xuefeng black-bone chicken intestinal morphology and microflora population. Our findings not only help solve the problems associated with the ban of antibiotics in livestock production, but also contribute to the development of new, effective alternatives to antibiotics, and provide a new scheme for improving intestinal health in Xuefeng black-bone chicken.

## Materials and methods

### Chickens, diet, and experimental design

For microbial fermentation of feed, the basal diet was mixed with probiotics, including *Bacillus subtilis* ($5 \times 10^8$ colony forming units/g), *Saccharomyces cerevisiae* ($5 \times 10^7$ colony forming units/g), *Lactobacillus plantarum* ($3 \times 10^8$ colony forming units/g), and *Enterococcus faecium* ($5 \times 10^8$ colony forming units/mL) (inoculum proportion = 1:1:3:3 v/v, inoculum size = 3% v/

v, moisture = 45% w/v, and temperature = 35 ± 1˚C). Additionally, molasses (2%) was added and mixed thoroughly using a feed-stuff mixer. The fermented feed was kept in a polyethylene bag with a one-way air valve during the fermentation process, and fermented by anaerobic fermentation for 20 days. The probiotic strains were obtained from the probiotics collection of Guangdong Institute Microbiology (Guangzhou, China).

Xuefeng black-bone chicken, a native breed of China, has excellent quality of meat with high contents of protein, vitamins, and amino acids, and hence was selected for this experiment. A total of 400 Xuefeng black-bone chickens (one-day-old and balanced for body weight) were used in a completely randomized design for a 7-d adaptation period and a 150-d experimental period. The chickens were randomly divided into 4 groups (4 replicates per group and 25 chickens in each replicate): 100% complete feed group (Cn group), 100% microbially fermented feed group (Fe group), 100% complete feed and ginseng polysaccharide (200 g/t) group (Po group), and 100% microbially fermented feed and ginseng polysaccharide group (FP group). In our paper, the Ginseng polysaccharides were extracted from the roots of ginseng with hot water, precipitated by 80% ethanol and deproteinated based on our previous method [12]. Their nutritional requirements were met according to the National Research Council feeding standards (NRC, 1994) and basic diet formulations and nutrient composition are shown in S1 Table. Chickens were raised in floor pens, and placed into separate floor pens with 20 individuals per pen. All birds received feed and fresh water *ad libitum* throughout the experiment. Feed was removed from the pen 24 hours before sampling. All animal procedures were approved by the Animal Care Committee at South China Agricultural University, and the experimental individuals were anaesthetized prior to exsanguination according to the University's guidelines for animal research.

## Data collection and sampling

On day 150, 2 broiler chickens were randomly selected from each pen (8 broiler chickens/treatment) and euthanized with an overdose of $CO_2$ [13]. The intestinal tract was immediately removed. Tissue samples of the jejunum were obtained and gently flushed with 0.9% saline to remove the intestinal contents and fixed in 10% formalin for histomorphological analysis. Subsequently, the jejunum contents (8 broiler chickens/treatment) were collected in sterile 1.5 mL tubes and stored at –80˚C until DNA isolation.

## Intestinal histomorphology

The intestinal samples were processed according to the method of Thompson et al. [14]. In brief, the intestinal samples were dehydrated with increasing concentrations of ethanol, cleared with xylene (Surgipath Medical Industries, Richmond, IL), and embedded in paraffin wax (Thermo Fisher Scientific, Kalamazoo, MC). Cross sections (5 μm) were stained with hematoxylin and eosin (GeneCopoeia, Rockville, MD). The stained sections were dehydrated with ethanol, cleared with xylene, and mounted with DPX mountant based on the method of Jiang et al. [15]. ImageJ software (National Institutes of Health, USA) was used to determine the morphometric measurements of villus height and crypt depth of the jejunum using an Olympus BX40 F-3 microscope (Olympus Cooperation, Tokyo, Japan) attached to a digital video camera (Q-imaging, 01-MBF-200R-CLR-12, SN: Q32316, Canada) [16].

## DNA extraction, 16S rRNA gene sequencing, and annotation analysis

Total genomic DNA from the jejunum contents was extracted using QIAamp DNA Stool Mini Kit (Qiagen, Hilden, Germany) according to the specifcations of the manufacturers. The DNA samples were tested for integrity using 1% agarose gel electrophoresis and their concentration

was determined using a Qubit fluorometer (Invitrogen, Carlsbad, CA). Based on the concentration, the DNA sample was diluted to 1 ng/µL using sterile water. The V3–V4 regions of the 16S ribosomal DNA genes were amplified by PCR based on the method of Sun et al. [17]. Sequencing libraries were generated using an Illumina MiSeq Reagent Kit (Illumina, San Diego, CA) on an Illumina MiSeq Sequencer. Single-end reads were assigned to samples based on their unique barcode in the adaptor sequence. Quality filtering of the raw reads was performed to obtain high-quality clean reads according to the Cutadapt quality controlling process [18]. The reads were compared with the reference database [19] using the UCHIME algorithm [20] to detect chimeric sequences [21], and clean reads were finally obtained using the Uparse software (Uparse v7.0.1001) [22]. Sequences with ≥ 97% similarity were assigned to the same operational taxonomic units (OTU). For each representative OTU, the Silva Database was used to annotate taxonomic information based on the Mothur algorithm [19]. To study the phylogenetic relationships between different OTU, multiple sequence alignment was performed using the MUSCLE software (Version 3.8.31) [23]. Alpha diversity was applied to analyze the complexity of species diversity within groups, including the Observed-species, Chao1, ACE, and Shannon indices. Beta diversity analysis was used to evaluate differences between groups using non-metric multi-dimensional scaling. Two different complementary analyses, analysis of similarity and multiresponse permutation procedure, were used to determine significant differences between jejunum microbiota in their response to fermented diets with or without ginseng polysaccharide. All these indices were calculated using the quantitative insights into microbial ecology pipeline (version 1.7.0) and displayed using the R software (version 2.15.3).

## Statistical analysis

The experimental design included a completely randomized design. All data, including villus height, crypt depth, and the ratio of villus height to crypt depth were analyzed using one-way ANOVA and Duncan's test with SPSS version 17.0 (IBM Corp., Armonk, NY). The R software was used to perform MetaStat analysis to determine differences in the relative abundance of fecal microbiomes [24]. The $P$ values indicated the significant differences at the levels of $P < 0.05$. The correlation analyses of microbiota composition with the intestinal morphology were tested by the cor function (x, y, use = "p") (http://127.0.0.1:11153/library/stats/html/cor.html), and illustrated with a function-labelled Heatmap (Matrix, xLabels, yLabels) in the R package WGCNA (http://127.0.0.1:11153/library/stats/html/cor.html).

## Results

### Histomorphological measurements of the intestine

The effects of fermented feed and ginseng polysaccharide on the villus height, crypt depth, and the ratio of villus height to crypt depth of the jejunum of Xuefeng black-bone chicken were evaluated, as shown in Fig 1. The villus height was significantly increased in the Fe and FP groups ($P < 0.05$) but not in the Po group compared to that in the Cn group ($P > 0.05$). The crypt depth was significantly decreased ($P < 0.05$) in the Fe, Po, and FP groups compared with that in the Cn group. However, no significant differences in crypt depth were observed between the Fe, Po, and FP groups ($P > 0.05$). The ratio of villus height to crypt depth was significantly higher ($P < 0.05$) in the Fe, Po, and FP groups compared with that in the Cn group. However, no differences in villus height to crypt depth ratio were observed between the Po and FP groups ($P > 0.05$) while higher than those in the Fe group ($P < 0.05$).

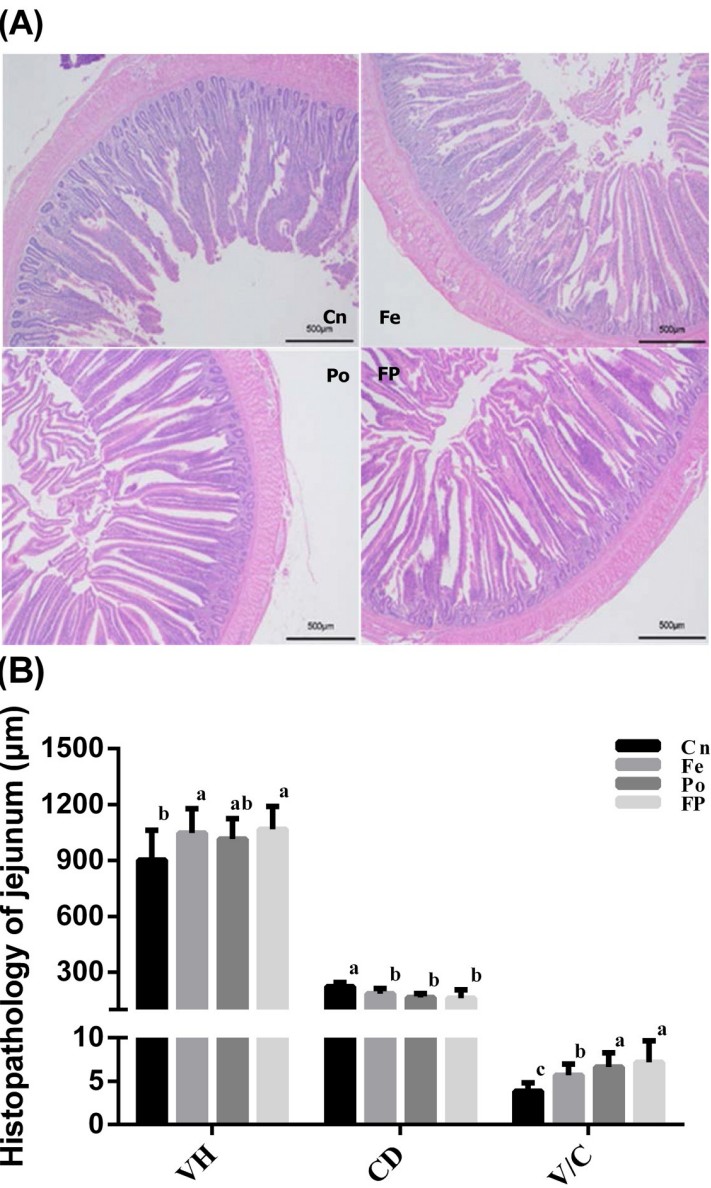

**Fig 1. Effects of dietary treatment on villus height, crypt depth, and the ratio of villus height to crypt depth in the jejunum of broiler chickens.** Treatments with different letters are significantly different at $P < 0.05$. Data were obtained from transmission electron microscopy, and were means of 10 birds (2 bird from each pen). Cn, 100% complete feed group; Fe, 100% microbial fermented feed group; Po, 100% complete feed and ginseng polysaccharide group; FP, 100% microbial fermented feed and ginseng polysaccharide group. VH, villus height; CD, crypt depth; V/C, the ratio of villus height to crypt depth.

## 16S rRNA gene sequencing and annotation analysis

After DNA extraction, the hypervariable V3–V4 regions of the 16S ribosomal DNA were enriched in each sample, and subsequent high-throughput analysis generated a total of 2,886,390 raw reads. On an average, each sample produced approximately 90,200 joined tags, which were assembled using PandaSeq v2.8 (min = 77,322, max = 99,325). Over 77.65% ±

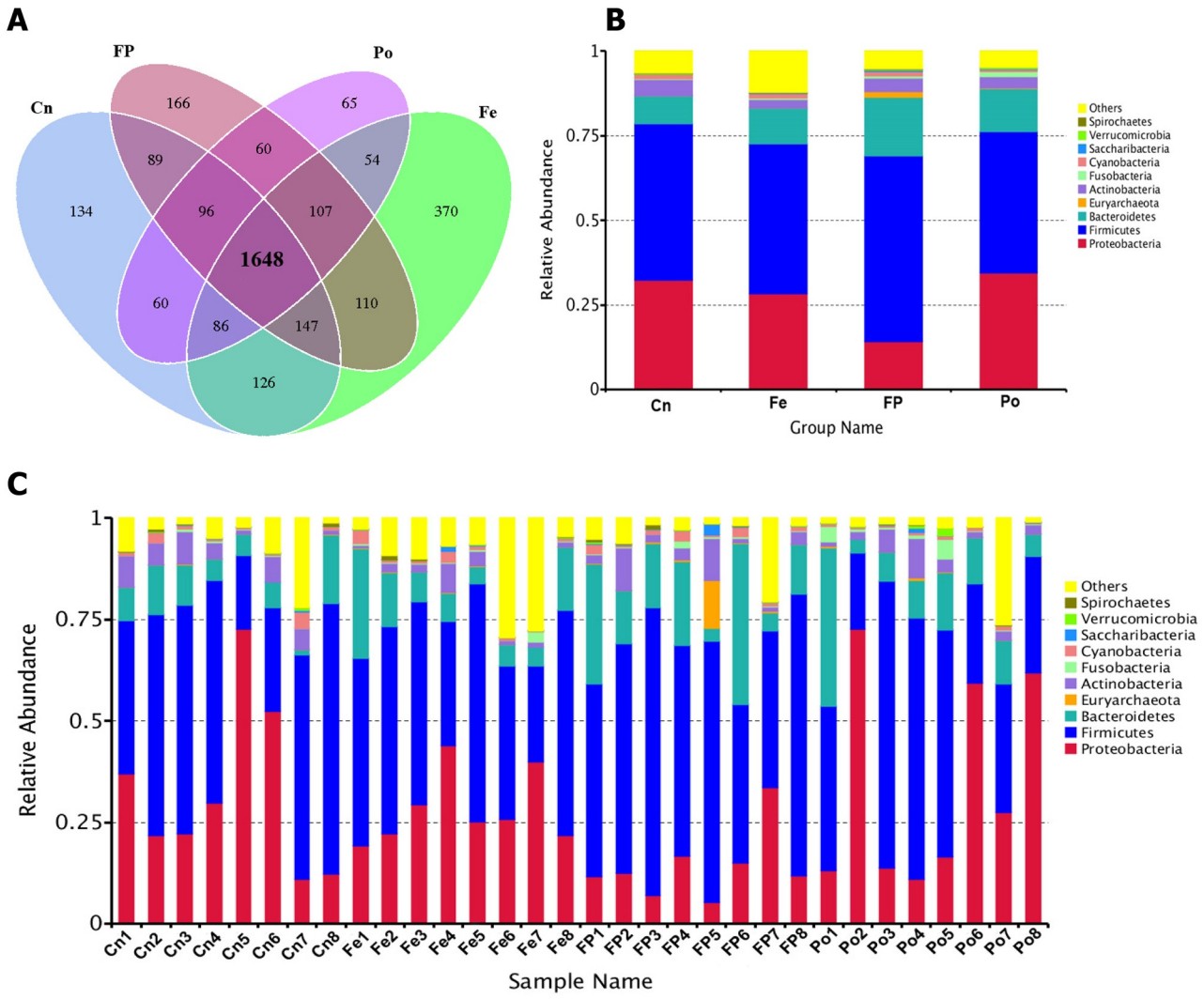

**Fig 2.** Venn Graph representation of the shared and exclusive OTUs at the 97% similarity level among the three groups of fecal microbiota (A). Bar plots analysis shows relative abundance of fecal microbiota at the phylum level in each group (B) and in each sample (C).

4.33% of the total joined tags from each sample passed quality control and were processed for further analysis (S2 Table). OTU clustering of the high-quality tags yielded 3,318 unique OTU candidates at 97% sequence similarity, and 1,648 candidates that were shared across all samples were defined as core OTU. The core OTU comprised approximately 49.67% of the total candidates, while only 134,166,65 and 370 OTU were identified uniquely in the Cn, FP, Po, and Fe groups (Fig 2A and S3 Table). Additionally, the microbial diversity in the jejunum contents of broiler chickens was assessed using quantitative insights into microbial ecology pipeline based on the OTU annotation, which identified the top 10 phyla (Fig 2B and 2C). The most abundant phylum in the jejunum contents of Xuefeng black-bone chicken was *Firmicute*s, which accounted for approximately 46.75% of all sequences, followed by *Proteobacteria* (27.36%) and *Bacteroidetes* (12.20%) (S4 and S5 Tables). At the class level, a total of 59 classes were detected, and 9 classes, including *Clostridia*, *Gammaproteobacteria*, *Bacteroidia*, *Epsilonproteobacteria*, *Bacilli*, *Alphaproteobacteria*, *unidentified_Actinobacteria*, *Deltaproteobacteria*,

and *Melainabacteria* had a relative abundance greater than 1.0%. The most abundant class in the jejunum contents of Xuefeng black-bone chicken was *Clostridia* (40.21%) (S6 and S7 Tables). At the order level, we detected 11 orders with a relative abundance greater than 1.0% (S8 and S9 Tables). Specifically, the most abundant order in the jejunum contents of Xuefeng black-bone chicken was *Clostridiales*, which accounted for approximately 39.73% of all sequences, followed by *Bacteroidales* (11.69%), *Campylobacterales* (7.12%), *Enterobacteriales* (6.17%), *Lactobacillales* (4.64%), *Oceanospirillales* (4.62%), *Alteromonadales* (1.87%), *Rhizobiales* (1.73%), *Desulfovibrionales* (1.53%), *Bifidobacteriales* (1.29%), and *Caulobacterales* (1.04%). At the family level, among the 189 families detected, 17 families, including *Ruminococcaceae*, *Lachnospiraceae*, *Helicobacteraceae*, *Enterobacteriaceae*, *Peptostreptococcaceae*, *Halomonadaceae*, *Bacteroidaceae*, *Lachnospiraceae*, *Rikenellaceae*, *Christensenellaceae*, *Shewanellaceae*, *Prevotellaceae*, *Desulfovibrionaceae*, *Clostridiales_vadinBB60_group*, *Bifidobacteriaceae*, *Caulobacteraceae*, and *Enterococcaceae* had a relative abundance greater than 1.0%. The most abundant family in the jejunum contents of Xuefeng black-bone chicken was *Ruminococcaceae* (19.23%) (S10 and S11 Tables). Among the 442 genera detected, 18 genera, including *Helicobacter*, *Serratia*, *Romboutsia*, *Bacteroides*, *Halomonas*, *Lactobacillus*, *Ruminococcaceae_UCG-010*, *Christensenellaceae_R-7_group*, *Rikenellaceae_RC9_gut_group*, *Shewanella*, *Lachnoclostridium*, *Ruminococcaceae_UCG-005*, *Enterococcus*, *Faecalibacterium*, *Eubacterium_coprostanoligenes_group*, *Desulfovibrio*, *Ruminococcaceae_UCG-014*, and *Ruminococcaceae_NK4A214_group* had a relative abundance greater than 1.0% (S12 and S13 Tables). At the species level, among the 216 species detected, 3 species, including *Serratia_marcescens*, *Bacteroides_barnesiae*, and *Shewanella_algae* had a relative abundance greater than 1.0%. The most abundant species in the jejunum contents of Xuefeng black-bone chicken was *Serratia_marcescens* (4.93%) (S14 and S15 Tables).

## Microbial diversity in the jejunum contents of Xuefeng black-bone chicken

We compared the alpha-diversity (within-sample diversity or estimate of species richness and evenness) of each sample with differing sequence counts or sampling efforts. Rarefaction curve analysis indicated that the number of sequences and sequencing depth were sufficient for this study (S1 Fig). We further used observed species, Chao1, ACE, and Shannon indices to evaluate the diversity of jejunum content of Xuefeng black-bone chicken among the different groups (Fig 3). Specifically, observed species, Chao 1, ACE, and the Shannon diversity index indicated a significant decrease in the diversity of jejunum content of the Po group compared with that of the Cn, Fe, and FP groups ($P = 0.007$, $P = 0.071$, $P = 0.003$, and $P = 0.003$, respectively). For beta diversity analysis, the relationship between the microbial communities in the jejunum of chickens fed different diets was assessed by non-metric multi-dimensional scaling, as shown in Fig 4. There were overlaps between the Cn and Fe groups, but the microbial communities of the FP and Po groups formed a distinct cluster separated from those of the Cn group. In addition, pairwise analysis of similarity [25] suggested that there were highly significant differences between the Cn and Fe groups (global R = 0.139, $P = 0.045$), the Cn and FP groups (global R = 0.239, $P = 0.004$), the Cn and Po groups (global R = 0.253, $P = 0.005$), the Fe and FP groups (global R = 0.387, $P = 0.003$), the Fe and Po groups (global R = 0.436, $P = 0.004$), and the FP and Po groups (global R = 0.165, $P = 0.05$) (Table 1). We also performed a multiresponse permutation procedure [26] in within- and between-group populations. The differences for between-group homogeneity were higher than those for within-group with an A value > 0, and the change degree reached a significant level between the Cn and Fe groups ($P = 0.031$), the Cn and FP groups ($P = 0.018$), the Cn and Po groups ($P = 0.01$), the Fe and FP

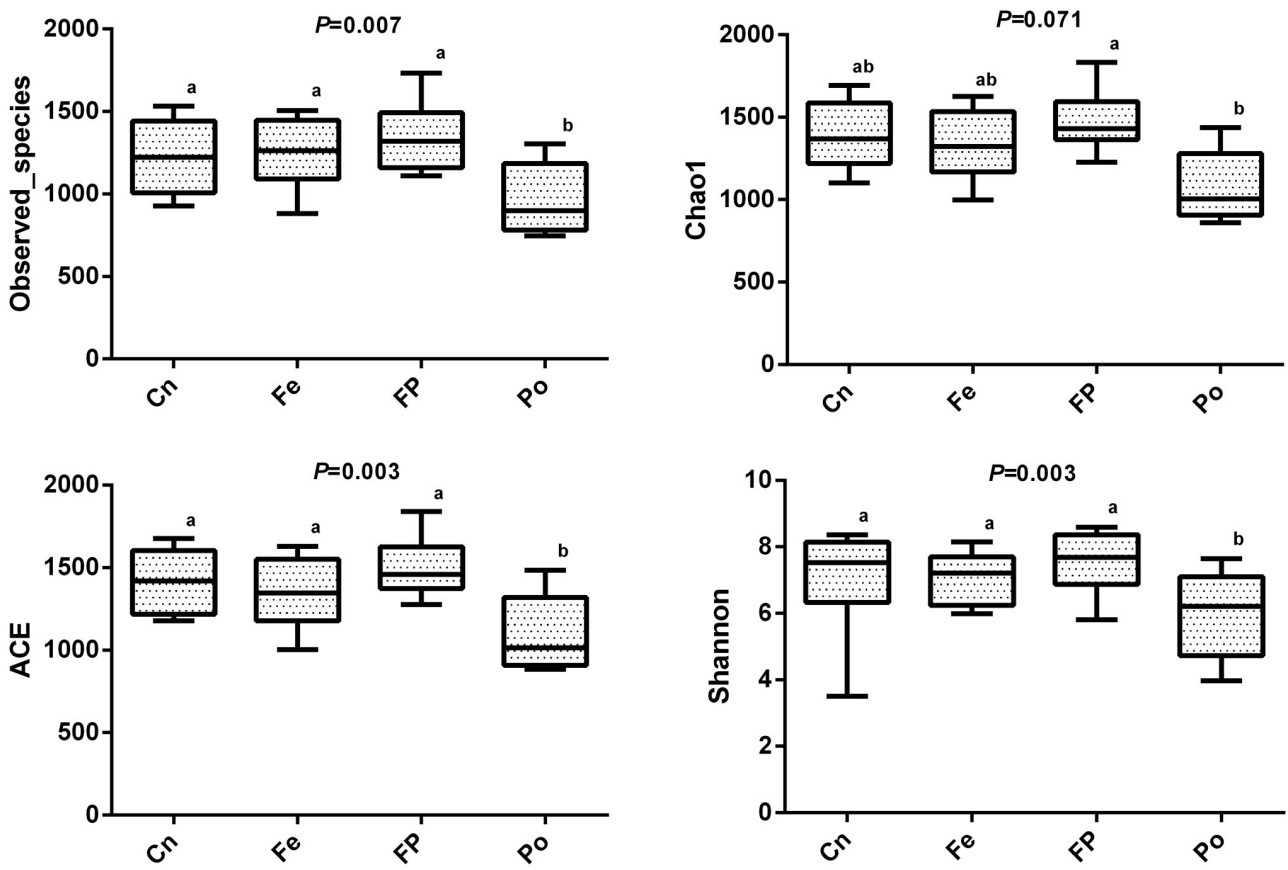

**Fig 3. Microbial diversity indices in the fecal microbiome.** Summary of four measures of α-diversity (observed species, Chao 1 index, ACE and Shannon).

groups ($P = 0.009$), the Fe and Po groups ($P = 0.004$), and the FP and Po groups ($P = 0.038$) (Table 1).

Anosim, analysis of similarity; MRPP, multiresponse permutation procedure. The R-value ranged from -1 to 1, and an R-value > 0 that showed the significant differences in the between-group population compared with those in the within-group population, as well as A-value (analysed by MRPP). The *P* values indicated the significant differences at the levels of $P < 0.05$ or $P < 0.01$.

## Alternative diets changed the intestinal microbiota composition

Differences in the microbial composition (relative abundance) of the jejunum microbiomes between the Cn and FP groups and the Cn and Po groups were evaluated by the MetaStat method using Fisher's exact test [24]. A total of 19 genera displayed a significant difference in relative abundance between the Cn and FP groups. Among them, the abundance of *Ureaplasma*, *Synechococcus*, *Coprococcus_1*, *Fonticella*, *Butyricicoccus*, *Elusimicrobium*, *Sutterella*, *Anaerostipes*, *Faecalitalea*, *Thalassospira*, and *Bacteroides* was increased, and that of *Sporolactobacillus*, *Bacillus*, *Halomonas*, *Aquamicrobium*, *Devosia*, *Microbacterium*, *Asteroleplasma*, and *Shewanella* was decreased in the FP group (S16 Table). Compared with that in the Cn group, the relative abundance of 4 genera, including *Thermoplasmatales*, *Odoribacter*,

# NMDS Analysis

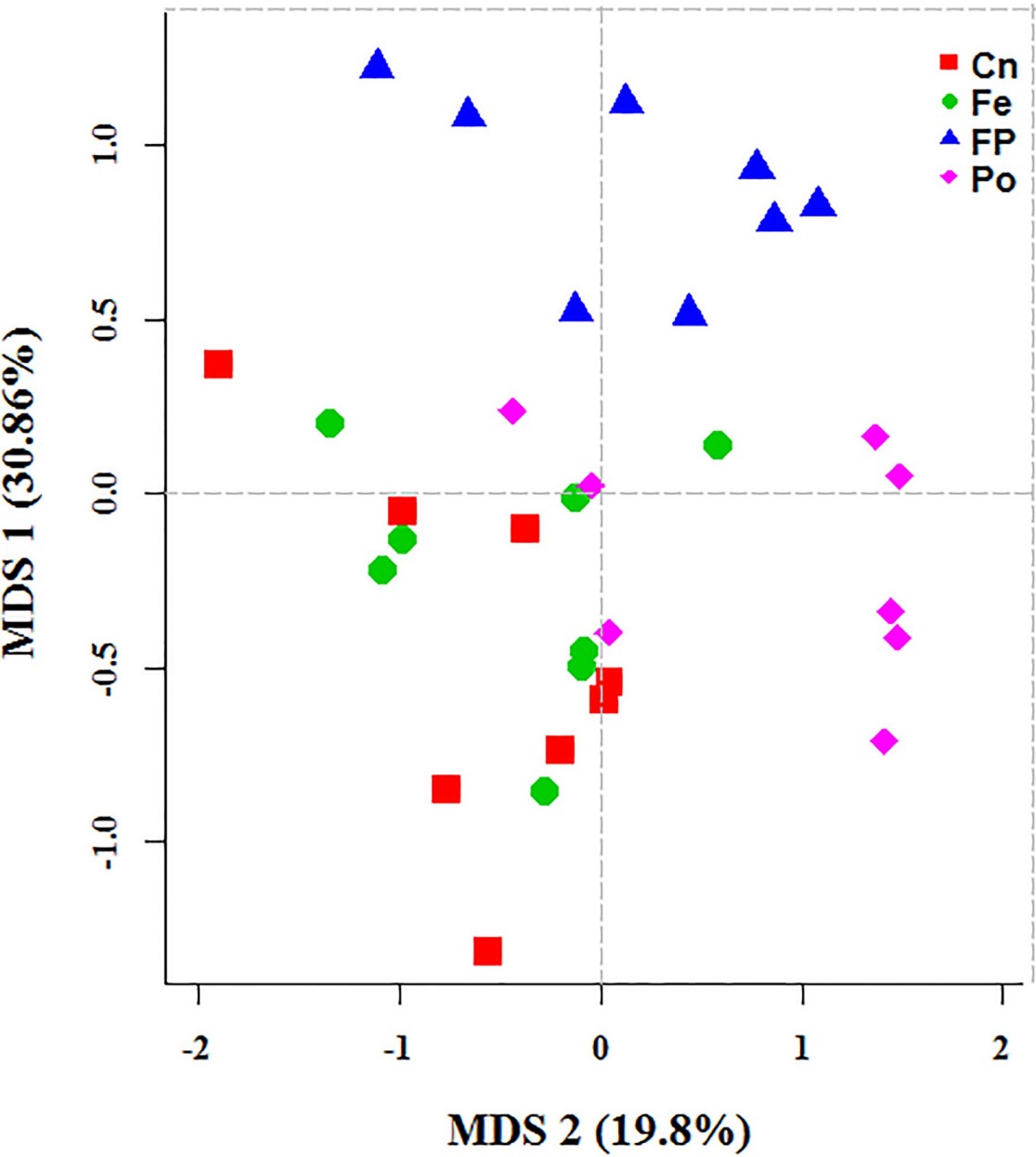

**Fig 4. First two dimensions from the (non-metric) multi-dimensional scaling (NMDS) of the Bray-Curtis dissimilarity matrix.**
Each point in the figure represents a sample, the distance between points indicates the degree of difference, and the samples in the same group are represented by the same color.

*Erysipelotrichaceae_UCG-003*, and *Sutterella* was significantly increased, and that of 7 genera, including *Bacillus*, *Coprococcus_2*, *Sporolactobacillus*, *Alkaliphilus*, *Aequorivita*, *Asterole-plasma*, and *Sphingobacterium* was significantly decreased in the Po group (S17 Table). At the species level, a total of 8 species displayed a significant difference in relative abundance

**Table 1. Significant differences in community structure in the jejunal microbiota of between different groups.**

| Group | Anosim | | MRPP | | | |
|---|---|---|---|---|---|---|
| | R value | P value | A value | Observed delta | Expected delta | P value |
| Cn-Fe | 0.139 | 0.045 | 0.024 | 0.607 | 0.622 | 0.031 |
| Cn-FP | 0.239 | 0.004 | 0.033 | 0.618 | 0.639 | 0.018 |
| Cn-Po | 0.253 | 0.005 | 0.036 | 0.632 | 0.655 | 0.01 |
| Fe-FP | 0.387 | 0.003 | 0.057 | 0.584 | 0.619 | 0.009 |
| Fe-Po | 0.436 | 0.004 | 0.069 | 0.597 | 0.642 | 0.004 |
| FP-Po | 0.165 | 0.05 | 0.028 | 0.608 | 0.626 | 0.038 |

between the Cn and FP groups. Among them, the abundance of *Clostridiales_bacterium_NK3B98*, *Lactobacillus_phage_Sal3*, *Bacteroides_vulgatus*, and *Eubacterium_tortuosum* was increased, and that of *Mycoplasma_gallinarum*, *Acinetobacter_lwoffii*, *Gallibacterium_anatis*, and *Asteroleplasma_anaerobium* was decreased (S18 Table). A total of 6 species displayed a significant difference in relative abundance between the Cn and Po groups. Among them, the abundance of *Methanogenic_archaeon_CH1270*, *Clostridiales_bacterium_60-7e*, and *Lactobacillus_phage_Sal3* was increased, and that of *Mycoplasma_gallinarum*, *Asteroleplasma_anaerobium*, and *Brevundimonas_diminuta* was decreased (S19 Table).

## Correlation between intestinal microbiota and histomorphology of jejunum

To further identify the genera that significantly correlated with the jejunum histomorphology, we performed Pearson's correlation test. Results showed that the villus height of the jejunum of Xuefeng black-bone chicken displayed a strong positive correlation with the relative abundance of *Coprococcus*_1, *Odoribacter*, and *Butyricicoccus* ($P = 0.02$, $P = 0.01$, and $P = 0.05$, respectively) and a negative correlation with that of *Coprococcus*_2 ($P = 0.008$) (Fig 5). The crypt depth of the

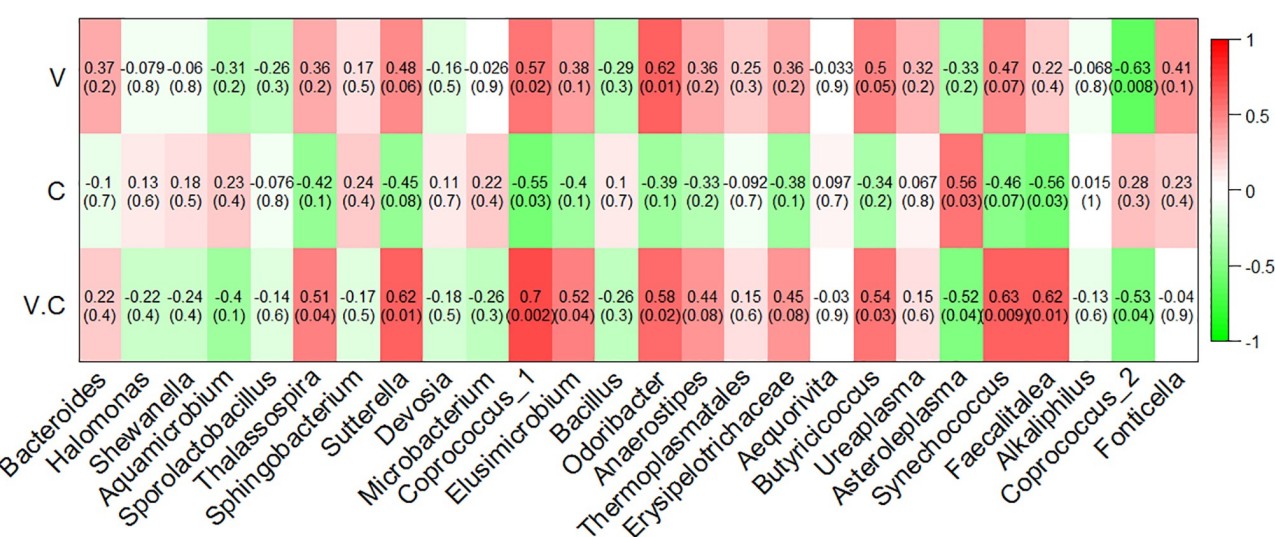

**Fig 5. Correlation analyses of genera taxa with the villus height, crypt depth, and the ratio of villus height to crypt depth in the jejunum of broiler chickens.** Each cell contains the corresponding correlation and *P*-value. The table is color-coded by correlation according to the color legend. V, the villus height; C, the crypt depth; V.C, the ratio of villus height to crypt depth.

jejunum of Xuefeng black-bone chicken showed a positive correlation with the relative abundance of *Asteroleplasma* ($P$ = 0.03), and a negative correlation with that of *Coprococcus_1* ($P$ = 0.03) and *Faecalitalea* ($P$ = 0.03) (Fig 5). The relative abundance of *Thalassospira*, *Sutterella*, *Coprococcus*_1, *Elusimicrobium*, *Odoribacter*, *Butyricicoccus*, *Synechococcus*, and *Faecalitalea* was positively correlated with the ratio of villus height to crypt depth of the jejunum of Xuefeng black-bone chicken ($P$ = 0.04, $P$ = 0.01, $P$ = 0.002, $P$ = 0.04, $P$ = 0.02, $P$ = 0.03, $P$ = 0.009, and $P$ = 0.01, respectively). However, the ratio of villus height to crypt depth was negatively correlated with the relative abundance of *Asteroleplasma* ($P$ = 0.04) and *Coprococcus_2* ($P$ = 0.04) (Fig 5). At the species level, the villus height of the jejunum of Xuefeng black-bone chicken showed a slight positive correlation with the relative abundance of *Clostridiales_bacterium*_60-7e ($P$ = 0.07) and a negative correlation with that of *Mycoplasma_gallinarum* ($P$ = 0.04) (S2 Fig). The crypt depth of the jejunum of Xuefeng black-bone chicken was positively correlated with the relative abundance of *Asteroleplasma_anaerobium* ($P$ = 0.03), and negatively correlated with that of *Bacteroides_vulgatus* ($P$ = 0.05) and *Eubacterium_tortuosum* ($P$ = 0.03) (S2 Fig). The ratio of villus height to crypt depth showed a positive correlation with the relative abundance of *Bacteroides_vulgatus* ($P$ = 0.01) and *Eubacterium_tortuosum* ($P$ = 0.01), and a negative correlation with that of *Asteroleplasma_anaerobium* ($P$ = 0.04) (S2 Fig).

## Discussion

The primary function of the small intestine is the digestion and absorption of nutrients by the intestinal mucosa [27]. Its function is closely related to the villus height, crypt depth, and villus height to crypt depth ratio. A previous study reported that changes in the intestinal morphology, such as increased villus height and villus height to crypt depth ratio, and shallow crypts indicate enhanced digestion and absorption in the small intestine [28]. Consistent with our findings, several studies have shown that dietary supplementation of fermented feed with probiotics significantly improves intestinal morphology [29], and increases the villus height, crypt depth, and villus height to crypt depth ratio in the jejunum of broilers [30, 31]. In this study, we demonstrated that the Fe group displayed significantly increased villus height and villus height to crypt depth ratio, and decreased crypt depth in the jejunum ($P$ < 0.05). This may be caused by fermentation of metabolites, including organic acids, amino acids, small peptides, and other substances, which improves the nutritional status of the intestine as well as intestinal morphology [32]. Additionally, we found that the Po group displayed slightly increased villus height, significantly increased villus height to crypt depth ratio, and significantly reduced crypt depth. This finding is in agreement with the results of Zahran et al. [32], who showed that dietary supplementation of astragalus polysaccharides caused a significant increase in the intestinal villus height in tilapia fish. Wang et al. [33] demonstrated that Sargassum fusiforme polysaccharides affect intestinal parameters, such as the ratio of villus height to crypt depth in mice. This may be caused by polysaccharides that are herbal plants with important biological activities to improve intestinal development [34]. Interestingly, in our study, compared to the Cn group, the FP group showed increased villus height, shallow crypt depth, and higher villus height to crypt depth ratio in the jejunum ($P$ < 0.05). However, there is currently no information on the effect of fermented feed and ginseng polysaccharides on the intestinal structure of broiler chickens, and hence, further research is required.

Next, we characterized the microbial composition of the intestinal contents of 32 Xuefeng black-bone chickens. A total of 2,245,803 high quality valid tags were obtained across all samples, and the sequence size of each sample ranged from 49,767 to 80,545, which was greater than that reported in previous studies on broilers [35]. The Chao1 index indicated

that this sequencing depth was sufficient for further analysis. Cluster analysis revealed that *Firmicute*s was the most dominant phyla among the total sequences. This finding is consistent with that of a study conducted by Wu et al. [36]. However, the functions of the phyla require further research. Additionally, in this study, the composition and structure of the gut microbial community was altered in the FP and Po groups. At the genus level, the FP and Po groups displayed significantly increased abundance of *Sutterella* and decreased abundance of *Asteroleplasma*. Interestingly, we found that the bacterial abundance of *Sutterella* was positively correlated to the villus height and villus height to crypt depth ratio, and negatively correlated to the crypt depth. In contrast, the bacterial abundance of *Asteroleplasma* was positively correlated to the crypt depth and negatively correlated to the villus height to crypt depth ratio. A previous study reported that the members of the genus *Sutterella* are widely prevalent in the human gastrointestinal tract, and their deficiency alters the colonic microbiota, resulting in disruption of the normal function of the colonic epithelium and inflammatory bowel disease [37, 38]. Min et al. [39] reported that an increase in the abundance of *Asteroleplasma* resulted in intestinal inflammation and other colonic epithelium diseases. At the species level, we discovered that the FP group showed significantly increased abundance of *Bacteroides_vulgatus* and *Eubacterium_tortuosum* and decreased abundance of *Mycoplasma_gallinarum* and *Asteroleplasma_anaerobium*. However, the abundance of *Mycoplasma_gallinarum* and *Asteroleplasma_anaerobium* was increased in the Po group, which may be because the combined effect of polysaccharides and fermented feeds is stronger than that of polysaccharides alone. Additionally, we found that the bacterial abundance of *Mycoplasma_gallinarum* was negatively correlated with the intestinal villus height. The crypt depth was positively correlated with the relative abundance of *Asteroleplasma_anaerobium*, and negatively correlated with that of *Bacteroides_vulgatus and Eubacterium_tortuosum*. However, the ratio of villus height to crypt depth was positively correlated with the relative abundance of *Bacteroides_vulgatus and Eubacterium_tortuosum*, and negatively correlated with that of *Asteroleplasma_anaerobium*. A previous study showed that an increase in the abundance of *Asteroleplasma_anaerobium* resulted in loss of pyruvate kinase activity, which slows glucose metabolism, thus hindering intestinal development, and an increase in the abundance of *Mycoplasma_gallinarum* resulted in intestinal inflammation and respiratory disease [40]. Previous studies have also reported that the relative abundance of *Bacteroides_vulgatus* is negatively correlated with inflammatory bowel disease [41]. Waidmann et al. [42] showed that an abundance of *Bacteroides_vulgatus* protected against *Escherichia coli*-induced colitis in gnotobiotic interleukin-2-deficient mice. Additionally, Tadayon et al. [43] reported that deficiency of *Eubacterium_tortuosum* caused suppurative inflammation of the intestine. Based on the above findings, we proposed a hypothesis that FP and Po might improve intestinal morphology by altering the microbial communities. However, further in-depth research is needed to confirm this hypothesis.

## Conclusions

The results of the present study indicated that the Fe, FP, or Po groups showed improved intestinal morphology, including increased villus height and villus height to crypt depth ratio, and decreased crypt depth of the jejunum of Xuefeng black-bone chicken. Additionally, we also compared the intestinal microbial composition of Xuefeng black-bone chicken among the FP and Po groups. Moreover, they are closely related to the histomorphological characteristics of chicken jejunum, including villus height, crypt depth, and villus height to crypt depth ratio. Therefore, a combination of ginseng polysaccharides and microbial

fermented feeds or ginseng polysaccharide alone can be used as a new, effective alternative to antibiotics for improving intestinal morphology. However, further in-depth research is required to reveal the underlying mechanisms.

## Supporting information

**S1 Fig. Rarefaction curves.** Number of sequences (A) and number of sample (B) rarefaction curves for the sampled jejunum microbiotas. Number of detected OTUs on the y-axis; number of sequences (A) and of samples (B) on the x-axis.
(PDF)

**S2 Fig. Correlation analyses of species taxa with the villus height, crypt depth, and the ratio of villus height to crypt depth in the jejunum of broiler chickens.**
(PDF)

**S1 Table. Basic diet formulations and nutrient composition.**
(DOCX)

**S2 Table. Quality control and preprocessing of metagenomic datasets.**
(XLSX)

**S3 Table. Sequence composition of each sample at each level in the Greengenes database.**
(XLSX)

**S4 Table. The aligned counts that were annotated at the phylum level.**
(XLSX)

**S5 Table. The aligned percentages that were annotated at the phylum level.**
(XLSX)

**S6 Table. The aligned counts that were annotated at the class level.**
(XLSX)

**S7 Table. The aligned percentages that were annotated at the class level.**
(XLSX)

**S8 Table. The aligned counts that were annotated at the order level.**
(XLSX)

**S9 Table. The aligned percentages that were annotated at the order level.**
(XLSX)

**S10 Table. The aligned counts that were annotated at the family level.**
(XLSX)

**S11 Table. The aligned percentages that were annotated at the family level.**
(XLSX)

**S12 Table. The aligned counts that were annotated at the genus level.**
(XLSX)

**S13 Table. The aligned percentages that were annotated at the genus level.**
(XLSX)

**S14 Table. The aligned counts that were annotated at the species level.**
(XLSX)

**S15 Table. The aligned percentages that were annotated at the species level.**
(XLSX)

**S16 Table. Significant differences between the Cn and FP groups identified at the genus-taxa level.**
(XLSX)

**S17 Table. Significant differences between the Cn and Po groups identified at the genus-taxa level.**
(XLSX)

**S18 Table. Significant differences between the Cn and FP groups identified at the species-taxa level.**
(XLSX)

**S19 Table. Significant differences between the Cn and Po groups identified at the species-taxa level.**
(XLSX)

## Author Contributions

**Data curation:** Jiajie Sun.

**Formal analysis:** Jiajie Sun.

**Investigation:** Jie Liu, Huan Wang.

**Project administration:** Junyi Luo, Ting Chen, Qianyun Xi, Yongliang Zhang, Jiajie Sun.

**Supervision:** Jiajie Sun.

**Writing – original draft:** Yueqin Xie, Jiajie Sun.

**Writing – review & editing:** Jiajie Sun.

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
