## [Decision Letter · Decision Letter 0]

3 Jul 2020

PONE-D-20-12303

Effects of fermented feeds and ginseng polysaccharides on the intestinal morphology and microbiota composition of Xuefeng black-bone chicken

PLOS ONE

Dear Dr. SUN,

Thank you for submitting your manuscript to PLOS ONE. After careful consideration, we feel that it has merit but does not fully meet PLOS ONE’s publication criteria as it currently stands. Therefore, we invite you to submit a revised version of the manuscript that addresses the points raised during the review process.

Both reviewers strongly urge the authors to provide much more detailed information in the Materials and Methods section.  Without an in-depth description of the experimental design, it was difficult to fully evaluated the validity of the data.  In your response letter, please provide a detailed description of the revisions made in the manuscript.  Since both reviewers thought that the data to be interesting and worthy of publication, please make a concerted effort to answer and respond to all comments of the reviewers.

We look forward to receiving your revised manuscript.

Kind regards,

Michael H. Kogut, Ph.D.

Academic Editor

PLOS ONE

Journal Requirements:

2. In your Methods, please state the method of anaesthesia used prior to exsanguination.

"This work was supported by grants from the Technical System of Modern Agricultural Industry in Guangdong Province [2018LM1121, 2018LM2158]."

"the Technical System of Modern Agricultural Industry in Guangdong Province [2018LM1121, 2018LM2158]."

Reviewers' comments:

Reviewer's Responses to Questions

**Comments to the Author**

1. Is the manuscript technically sound, and do the data support the conclusions?

Reviewer #1: Yes

Reviewer #2: Yes

2. Has the statistical analysis been performed appropriately and rigorously? 

Reviewer #1: I Don't Know

Reviewer #2: Yes

3. Have the authors made all data underlying the findings in their manuscript fully available?

Reviewer #1: Yes

Reviewer #2: Yes

4. Is the manuscript presented in an intelligible fashion and written in standard English?

Reviewer #1: Yes

Reviewer #2: Yes

5. Review Comments to the Author

Reviewer #1: The experiments seems well done and the data is important. However, the paper must have better description of Material and methods and stats.

The main topic, polisaccharides should be better addressed in the introduction, methods (describing better the product used) and discussion.

Also, the authors must to discuss better the data and correlate their findings with the literature, focusing on chickens.

Further comments are described in the file attached.

Reviewer #2: Abstract – group names were not defined before using abbreviations. The statement that basal diets were fermented by a variety of bacteria doesn’t make sense. The diet was actually fermented, or do you mean supplemented with these bacteria?

Introduction – Define microbially fermented feed, most readers will not know what this is.

Line 122 – was feed refreshed at these times and offered ad libitum or were birds fed a designated amount?

Was fermented feed the only source of nutrition for the birds, or was it combined with normal feed? If combined, at what ratio?

Was a nutrient composition analysis completed on the two different feeds? Perhaps differences noted in the experimental results were a result of changes in composition of the feed, and this would be interesting to note.

Line 116 describes 5 replicate pens per group, but line 130 suggests 4 replicate pens per group

In the discussion section, anywhere you refer to data shown in tables and figures, you should reference them.

The method of determining correlation between microbiome and intestinal morphometry is missing from the M&M section.

Mycoplasma gallinarum is a pathogen of chickens, they did not show clinical signs of disease despite the high abundance and presence in birds? This would explain the association with decreased villus height.

6. PLOS authors have the option to publish the peer review history of their article (what does this mean?). If published, this will include your full peer review and any attached files.

Reviewer #1: No

Reviewer #2: **Yes: **LR Bielke

---

## [Author Response · Author response to Decision Letter 0]

21 Jul 2020

Journal Requirements:

Answer: Many thanks for your kind help, and we have regulated to meet PLOS ONE's style requirements.

2. In your Methods, please state the method of anaesthesia used prior to exsanguination.

Answer: we have detailed the method of anaesthesia in main text with new line 131-133. The details as follow: On day 150, 2 broiler chickens were randomly selected from each pen (8 broiler chickens/treatment) and euthanized with an overdose of CO2 [12]. The intestinal tract was immediately removed.

"This work was supported by grants from the Technical System of Modern Agricultural Industry in Guangdong Province [2018LM1121, 2018LM2158]."

"the Technical System of Modern Agricultural Industry in Guangdong Province [2018LM1121, 2018LM2158]."

Answer: Thanks. We have removed the Acknowledgments section of our manuscript.

Reviewers' comments:

Reviewer's Responses to Questions

Comments to the Author

1. Is the manuscript technically sound, and do the data support the conclusions?

Reviewer #1: Yes

Reviewer #2: Yes

Answer: Thanks.

2. Has the statistical analysis been performed appropriately and rigorously?

Reviewer #1: I Don't Know

Reviewer #2: Yes

Answer: Thanks.

3. Have the authors made all data underlying the findings in their manuscript fully available?

Reviewer #1: Yes

Reviewer #2: Yes

Answer: Thanks.

4. Is the manuscript presented in an intelligible fashion and written in standard English?

Reviewer #1: Yes

Reviewer #2: Yes

Answer: Thanks.

5. Review Comments to the Author

Reviewer #1: 

The experiments seems well done and the data is important. However, the paper must have better description of Material and methods and stats. The main topic, polysaccharides should be better addressed in the introduction, methods (describing better the product used) and discussion. Also, the authors must to discuss better the data and correlate their findings with the literature, focusing on chickens. Further comments are described in the file attached.

Ethics Statement- Number of approve

ABSTRACT

Abreviation of the groups are not explained 

Answer: Thanks so much. The informations of groups were showed in M&M section with new line 113-116. And we have explained the abreviation of the groups in our Abstract with red words.

INTRODUTION 

Were all the diets fermented? No control diet? Distinguish Fe FP and Po groups

Answer: The related information was showed in M&M section with new lines 111-117. In our paper, a total of 400 Xuefeng black-bone chickens were randomly divided into 4 groups (5 replicates per group and 20 chickens in each replicate): 100% complete feed group (Cn group), 100% microbially fermented feed group (Fe group), 100% complete feed and ginseng polysaccharide (200 g/t) group (Po group), and 100% microbially fermented feed and ginseng polysaccharide group (FP group).

MATERIAL AND METHODS

Composition table of “basal diet” is missing- It is not complemental information 

Need a better explanation of complete feed. Is that the basal diet without fermentation?

Describe phases of diet

Answer: Thanks so much for professional suggestions. In our paper, basic diet formulations and nutrient composition was showed in S1 Table. In our experiment, the basal diet was not fermented and termed as control group. Two phases of diets were described in our paper, and the information was also showed in S1 Table.

Describe how the animals were raised (installation, building, shavings, etc)

Answer: Thanks so much. And we have re-prepared the text with new lines 122-125. The details were also following as “Chickens were raised in floor pens, and placed into separate floor pens with 20 individuals per pen. Feed was provided twice a day at 8:00 am and 5:00 pm, and drinking water was provided ad libitum throughout the experimental period. Feed was removed from the pen 24 hours before sampling.”

“cetyltrimethylammonium bromide/sodium dodecyl sulfate method” – describe method or add reference

Answer: we have revised in the main text with lins 155-157. The details as follow: Total genomic DNA from the jejunum contents was extracted using QIAamp DNA Stool Mini Kit (Qiagen, Hilden, Germany) according to the specifcations of the manufacturers.

Composition of the ginseng polyssacaride- describe better the product. Is it from any kind of extraction? What its composition? 

Answer: Thanks so much for your suggestion. Ginseng, one of the most well-known oriental medicine for several thousand years, has been widely used with mysterious powers as a tonic, prophylactic and restorative agent in china. In general, ginseng polysaccharides have a large content of polysaccharides, peptides, saponins, and other active substances. In our paper, the Ginseng polysaccharides were extracted from the roots of ginseng with hot water, precipitated by 80% ethanol and deproteinated based on our previous method. We have detailed in our text with lines 118-120.

Statistical Analysis- What P-value was considered significant? 

-Was the data tested to normality? What was made with non normal data? 

Answer: Thanks so much. We have added the related information in our main text with new lines 189-190. The details also showed as “The P values indicated the significant differences at the levels of P < 0.05.” Yes, the tested data was normal in our paper. Thanks again.

RESULTS

Figure 1. Describe the abbreviation of the graphics

Answer: Thanks so much. We have describe the abbreviation of the graphics. The details as bellow.

Fig 1. Effects of dietary treatment on villus height, crypt depth, and the ratio of villus height to crypt depth in the jejunum of broiler chickens. Treatments with different letters are significantly different at P < 0.05. Data were obtained from transmission electron microscopy, and were means of 10 birds (2 bird from each pen). Cn, 100% complete feed group; Fe, 100% microbial fermented feed group; Po, 100% complete feed and ginseng polysaccharide group; FP, 100% microbial fermented feed and ginseng polysaccharide group. VH, villus height; CD, crypt depth; V/C, the ratio of villus height to crypt depth.

Do you have the bromatological analysis of the feeds? Especially the fermented ones?

Would be interesting have the analysis of organic acids. 

Answer: Thanks so much for your professional suggestion. In our paper, basic diet formulations and nutrient composition was analyzed and presented in S1 Tables, while the other characteristics of feeds, such as organic acids, were not tested. And we very interested to do these analysis, and thanks so much for your professional advices. Now, it is a pity that we have no more feed samples. In the future, we will follow your opinions to improve our research. Thanks.

Line 354- What kinds of polysaccharides?

You should explain better which kinds of polisaccharides are you talking. .

Line 355- 358- how polysaccharide have protein, minerals, …?

Answer: Thanks so much. In line 354 (new line 364), the reference 33 illustrated Sargassum fusiforme polysaccharides, and we have added related information in our paper in new line 363-364. In new line 365-366, we also revised “This may be caused by polysaccharides that are not only herbal plants with important biological activities but also contain abundant proteins, carbohydrates, vitamins, fats, and minerals, which play an important role in maintaining nutrient balance, thus improving intestinal development” to “This may be caused by polysaccharides that are herbal plants with important biological activities to improve intestinal development”.

Please describe correlations of villus height and crypt deep with chicken microbiota already find in the literature! Also, discuss data on microbiota and performance in chickens, how that can be associated with your data?

Answer: Many thanks for your kind helps. We have details the methods of correlation analysis in the M&M section with new lines 190-194. The details were also showed as bellow: The correlation analyses of microbiota composition with the intestinal morphology were tested by the cor function (x, y, use = “p”) (http://127.0.0.1:11153/library/stats/html/cor.html), and illustrated with a function-labelled Heatmap (Matrix, xLabels, yLabels) in the R package WGCNA (http://127.0.0.1:11153/library/stats/html/cor.html). And we also try our best to discuss our data between microbiota and performances. Thanks so much.

Figure and table subtitle should be better described.

Answer: Thanks. We have detailed the subtitle of our Figures and tables.

Reviewer #2:

Abstract – group names were not defined before using abbreviations. The statement that basal diets were fermented by a variety of bacteria doesn’t make sense. The diet was actually fermented, or do you mean supplemented with these bacteria?

Answer: Thanks so much. We have detailed the abreviation of the groups in our Abstract with red words. And the fermented methods were showed in M&M section with the lines 99-108. 

Introduction – Define microbially fermented feed, most readers will not know what this is.

Answer: In the Introduction, we have added some description about fermented feed in new lines 71-73. The details were also showed as follow: “In general, the basal diet that fermented with probiotics has been extensively studied due to its benefits of increasing nutrient bioavailability and nutritional value [4]”.

Line 122 – was feed refreshed at these times and offered ad libitum or were birds fed a designated amount?

Was fermented feed the only source of nutrition for the birds, or was it combined with normal feed? If combined, at what ratio?

Was a nutrient composition analysis completed on the two different feeds? Perhaps differences noted in the experimental results were a result of changes in composition of the feed, and this would be interesting to note.

Answer: Thanks so much for your professional suggestion. In our paper, All birds received feed and fresh water ad libitum throughout the experiment. Feed was removed from the pen 24 hours before sampling. And we have also revised in our main text with new lines 123-125. The fermented feed was only for Fe (100% microbially fermented feed group) and FP (100% microbially fermented feed and ginseng polysaccharide group) groups. The information was described in new lines 111-117. In our paper, basic diet formulations and nutrient composition was analyzed and presented in S1 Tables, while the characteristics of fermented feed were not tested. And we very interested to do these analysis, but it is a pity that we have no more feed samples. In the future, we will follow your opinions to improve our research. Thanks.

Line 116 describes 5 replicate pens per group, but line 130 suggests 4 replicate pens per group

In the discussion section, anywhere you refer to data shown in tables and figures, you should reference them.

The method of determining correlation between microbiome and intestinal morphometry is missing from the M&M section.

Mycoplasma gallinarum is a pathogen of chickens, they did not show clinical signs of disease despite the high abundance and presence in birds? This would explain the association with decreased villus height.

Answer: Thanks so much for your kind help, and we have corrected in new line 113-114 as “The chickens were randomly divided into 4 groups (4 replicates per group and 25 chickens in each replicate)”. We have added the methods in new lines 189-194 as “The correlation analyses of microbiota composition with the intestinal morphology were tested by the cor function (x, y, use = “p”) (http://127.0.0.1:11153/library/stats/html/cor.html), and illustrated with a function-labelled Heatmap (Matrix, xLabels, yLabels) in the R package WGCNA (http://127.0.0.1:11153/library/stats/html/cor.html)”. Thanks so much for your professional suggestion about Mycoplasma gallinarum, and we have also updated our discussion. 

6. PLOS authors have the option to publish the peer review history of their article (what does this mean?). If published, this will include your full peer review and any attached files.

Do you want your identity to be public for this peer review? For information about this choice, including consent withdrawal, please see our Privacy Policy.

Reviewer #1: No

Reviewer #2: Yes: LR Bielke

Answer: Thanks.

---

## [Editor Report · Decision Letter 1]

24 Jul 2020

Effects of fermented feeds and ginseng polysaccharides on the intestinal morphology and microbiota composition of Xuefeng black-bone chicken

PONE-D-20-12303R1

Dear Dr. SUN,

We’re pleased to inform you that your manuscript has been judged scientifically suitable for publication and will be formally accepted for publication once it meets all outstanding technical requirements.

Kind regards,

Michael H. Kogut, Ph.D.

Academic Editor

PLOS ONE
---

## [Editor Report · Acceptance letter]

30 Jul 2020

PONE-D-20-12303R1 

Effects of fermented feeds and ginseng polysaccharides on the intestinal morphology and microbiota composition of Xuefeng black-bone chicken 

Dear Dr. SUN:

I'm pleased to inform you that your manuscript has been deemed suitable for publication in PLOS ONE. Congratulations! Your manuscript is now with our production department. 

Kind regards, 

on behalf of

Dr. Michael H. Kogut 

Academic Editor

PLOS ONE